# Effect of Radial-Shear Rolling on the Structure and Hardening of an Al–8%Zn–3.3%Mg–0.8%Ca–1.1%Fe Alloy Manufactured by Electromagnetic Casting

**DOI:** 10.3390/ma16020677

**Published:** 2023-01-10

**Authors:** Yury V. Gamin, Nikolay A. Belov, Torgom K. Akopyan, Victor N. Timofeev, Stanislav O. Cherkasov, Mikhail M. Motkov

**Affiliations:** 1Department of Metal Forming, National University of Science and Technology «MISIS» (NUST MISIS), 4 Leninsky Pr., 119049 Moscow, Russia; 2Department of Materials Science, Moscow Polytechnic University, 38, Bolshaya Semyonovskaya Str., 107023 Moscow, Russia; 3Electrotechnical Department, Siberian Federal University, 79 Svobodny Pr., 660041 Krasnoyarsk, Russia

**Keywords:** aluminum alloy, Al–Zn–Mg–Ca–Fe system, electromagnetic casting, FEM simulation, radial shear rolling, stress–strain state

## Abstract

Aluminum alloys are one of the most common structural materials. To improve the mechanical properties, an alloy of the Al–Zn–Mg–Ca–Fe system was proposed. In this alloy, when Fe and Ca are added, compact particles of the Al_10_CaFe_2_ compound are formed, which significantly reduces the negative effect of Fe on the mechanical properties. Because of the high solidification rate (about 600 K/s) during cylindrical ingot (~33 mm) production, the electromagnetic casting method (ECM) makes it possible to obtain a highly dispersed structure in the cast state. The size of the dendritic cell is ~7 μm, while the entire amount of Fe is bound into eutectic inclusions of the Al_10_CaFe_2_ phase with an average size of less than 3 μm. In this study, the effect of radial shear rolling (RSR) on the formation of the structure and hardening of the Al–8%Zn–3.3%Mg–0.8%Ca–1.1%Fe alloy obtained by EMC was studied. Computer simulation of the RSR process made it possible to analyze the temperature and stress–strain state of the alloy and to select the optimal rolling modes. It was shown that the flow features during RSR and the severe shear strains near the surface of the rod (10 mm) provided a refining and decrease in the size of the initial Fe-containing particles.

## 1. Introduction

Aluminum alloys are one of the most common structural materials. This is due to the highest content of Al in the Earth’s crust being found among all metals, as well as its possibility of achieving the most balanced set of basic service and technological properties [1,2,3]. However, there is a problem associated with the need to significantly improve the strength properties of aluminum alloys at a sufficiently high allowable concentration of iron (preferably at least 1%), as this element is present in technical grades of primary aluminum and especially in secondary raw materials [4].

Additives of Zn and Mg (as well as Cu in grade alloys) make it possible to achieve the highest strength properties (tensile strength up to 700 MPa [5]), which are realized in grade wrought alloys of the 7xxx series type (type 7075/7150) [1,6,7,8]. The disadvantage of these alloys is the strict requirement for purity in terms of impurities, primarily Fe, which does not allow for preparing such alloys from cheap charge materials [9,10]. On the other hand, it is known that at elevated cooling rates, dispersion of the cast structure, including Fe-containing phases, occurs [11].

To achieve high solidification rates in relation to the production of long workpieces, the Magnetic Hydrodynamics Scientific and Production Center LLC proposed and tested the electromagnetic casting technology (EMC) [12,13]. Pilot-plant equipment for the production of aluminum alloys through casting in EMC was developed, patented, and implemented, registered under the ElmaCast trademark (www.elmacast.com, accessed on 25 November 2022). ElmaCast technology provides cast billet cooling rates comparable to those of RS/PM technology (over 1000 K/s for billet diameters up to 14 mm). It was also successfully tested in industrial conditions for experimental aluminum alloys, in particular with additives of rare earth metals [14,15], namely iron, calcium, and zirconium [16,17]. This suggests that the application of the EMC method to high-strength wrought aluminum alloys based on the Al–Zn–Mg system with a high Fe content is promising.

It was shown in [16,18] that with the joint introduction of Fe and Ca into aluminum alloys, it is possible to bind iron into compact particles of the Al_10_CaFe_2_ compound instead of needle-shaped inclusions of the Al_3_Fe phase. As a consequence, a significant decrease in the negative effect of this element on the mechanical properties was observed in both castings and deformed semifinished products obtained from alloys of the Al–Zn–Mg–Ca–Fe system [19,20]. Hence, the expediency of considering such alloys in relation to the ElmaCast technology occurred.

In addition to the option of adding alloying elements to the composition of the alloy and the method of smelting, the method of deformation affects the final mechanical properties. Radial shear rolling (RSR) is an effective method for structuring materials. Radial shear rolling is a variant of helical rolling, which is implemented on three-roll rolling mills, where the rolls have a feed angle of more than 18 degrees [21,22]. Because of this, severe shear strains occur, which provide severe deformation of the material, thus significantly increasing its properties [23]. The authors of [24,25] showed the effectiveness of RSR in relation to aluminum alloys. Gamin et al. [26] studied the influence of the temperature–velocity parameters of radial-shear rolling (RSR) on the structure and properties of the A1050 (Al 99.5%) aluminum alloy using finite element modeling. It was shown that the strength of the obtained RSR rods in all modes was significantly higher than the strength of the industrial rods in a hot-pressed condition. Valeev et al. [27], Galkin et al. [28], and Mashekov et al. [29] investigated the formation of the structure and properties of bars made from aluminum alloy D16, obtained by the RSR method. The results showed that radial shear rolling has a positive effect on improving the material properties.

Previous data suggest that the processing of an ingot obtained by the EMC method with the combination of RSR will provide a significant effect and a higher level of mechanical properties.

Considering the above, the aim of this work is to study the main stress–strain parameters of the RSR process on the structure evolution of the experimental alloy of Al–Zn–Mg–Ca–Fe system manufactured through electromagnetic casting. To analyze the stress–strain state and select the optimal rolling modes, it is proposed to carry out the simulation of RSR using the finite element method.

## 2. Materials and Methods

### 2.1. Experimental Procedure

To carry out the study, the experimental alloy ZCF was used; the nominal and actual composition of the alloy is given in Table 1. The alloy was produced in an induction furnace based on technical purity aluminum (99.85%). Zinc and magnesium were introduced into the aluminum melt in the form of pure metals (99.9% purity), whereas iron and calcium were introduced in the form of binary master alloys (Al–10%Fe and Al–10%Ca, respectively).

The billet, with a length of 120 mm, was cut from an ingot with a diameter of ~33 mm, obtained by the ElmaCast technology, and was then used for the RSR process. RSR was carried out using the rolling mill “MISIS 14–40” (Figure 1a) in four passes to a final diameter of 10 mm at 450 °C (preliminary heating in the furnace for 3 h). The schematic diagram of RSR is shown in Figure 1b. The three rolls were symmetrically located relative to the rolling axis, and were rotated by the feed angle *β* and the rolling angle *γ*. During the rolling process, the rolls rotated in one direction and allowed the billet to move along a helical trajectory.

The rolling modes, including the elongation ratio after each pass, are presented in Table 2. The total elongation ratio Σ*µ* (that is, the ratio of the cross-sectional area of the initial billet to the area of the final rod) after four passes was equal to 10.5. The ingot and rod (the final length of the rod reaches ~1400 mm) are shown in Figure 2. Annealing and aging of the samples were carried out in a muffle electric furnace SNOL 8.2 and an oven SNOL 3.5.3.5.3.5/3.5-I1M, respectively.

### 2.2. Microstructure and Hardness Measurement

The microstructure was examined using transmission electron microscopy (TEM, Jeol JEM 1400 microscope) and scanning electron microscopy (SEM, TESCAN VEGA 3) with an electron microprobe analysis system (EMPA, OXFORD Aztec, Oxford Instruments, Oxfordshire, UK) and Aztec software. Polished samples were used for the studies. The metallographic samples were ground with SiC abrasive paper and polished with 1 µm diamond suspension. A 1% hydrogen fluoride (HF) water solution was used for the etching. Thin foils for TEM were prepared by two-jet electrolytic polishing on a STRUERS Tenupol-5 installation at 21 V voltage. A solution of nitric acid in methanol in a ratio of 1:4 (vol.) cooled to –15 °C was used as an electrolyte.

Vickers’ microhardness was measured using a Metkon Duroline MH-6 universal tester at a load of 100 g and a dwell time of 10 s. The measurements were carried out at least five times at each point. The standard deviation doesn’t exceed 3 HV.

Thermo-Calc (TTAL5 database) [30] was used to calculate the phase composition of the experimental alloy. As the database used did not contain a description of the compound Al_10_CaFe_2_, the calculation results were corrected, taking into account the experimental data.

### 2.3. FEM Simulation

Computer simulation was performed in the QForm 3D software package. The three-dimensional model included three rolls and the initial billet and corresponded to the scheme of RSR, as shown in Figure 1b.

The calculation is based on the finite element method. The billet and tool are divided into a mesh of finite elements (tetrahedrons). During the simulation process, the thermal fields in the billet and tool are calculated, taking into account the contact interaction (heat transfer) and convection heat exchange with the environment. In this case, the temperature of the tool and billet changes in accordance with the equation of transient heat conduction, accounting for heat transfer in the environment on the free surface (1):(1)$$q_{n}=b\alpha \left(T_{1}-T_{2}\right),$$
where *q_n_* is the heat-transfer rate through the heat transfer surface, W/m^2^; *α* is the heat transfer coefficient (W/K·m^2^), which takes into account the complex coefficients of heat transfer between the billet and lubricant and between the lubricant and tool; *T_1_* is the billet temperature, K; *T_2_* is the tool temperature, K; and *b* = 0.05 is the pause coefficient, which shows how many times it is necessary to reduce a heat transfer rate in the absence of tight contact between the billet and tool (without billet deformation).

Heat transfer by thermal conductivity is carried out through the direct contact of body parts with different temperatures. The differential equation of non-stationary heat conduction used for modeling (2) is as follows:(2)$$k\left(\Delta T\right)+q_{G}=\rho c\dot{T},$$
where *k* is the thermal conductivity, W/mK; Δ*T* is the difference of temperature field, K; *ρ* is the density kg/m^3^; *c* is the specific heat, J/kgK; and *q_G_* is the power of the internal heat source, W/m^3^.

Convective heat transfer occurs when the surface of a solid body makes contact with a gas (or liquid) at a different temperature. The QForm 10 program uses the Newton–Richmann law to describe convective heat transfer (3):(3)$$q_{n}=h\left(T_{1}-T_{c}\right),$$
where *h* is the heat transfer coefficient W/m^2^K; *T*_1_ is the body temperature (billet or tool); and *T_C_* is an ambient temperature.

The temperature of the billet or rod before each pass is set to be uniform and equal to 450 °C, the rolls’ temperature is equal to 50 °C, and the ambient temperature is equal to 20 °C.

In addition, based on the rheological properties of the workpiece material, the stress–strain state is calculated (Figure 3). To describe the rheological properties of the Al–Zn–Mg–Ca–Fe alloy, experimental data were used [31]. The flow stress is calculated as a function of temperature *T*, strain rate $\dot{\varepsilon }$, and deformation *ε* (4):(4)$$\sigma_{s}=f\left(T,\;\dot{\varepsilon },\varepsilon \right).$$

The initial billet for the simulation was a cylinder with a diameter of 33 mm and a length of 150 mm. Rolling was carried out in four passes to a final diameter of 10 mm, similar to the modes in Table 2. The elongation ratio in each pass *µ*_i_ can be determined as follows (5):(5)$$\mu_{i}={\left(\frac{D_{i-1}}{D_{i}}\right)}^{2}.$$
where *D_i_*_−1_ is the diameter of the billet before the *i* pass, and *D_i_* is the diameter of the rod after the pass.

In the simulation Siebel’s law was used to describe the contact friction between the billet and the tool. According to this law, the specific friction force is proportional to the maximum shear stress *k* with a proportionality factor called the friction factor *m* (6):(6)τ=mk.

The maximum shear stress *k* is determined by the resistance to deformation of the material *σ_s_* (7):(7)$$k=\frac{\sigma_{s}}{\sqrt{3}}.$$

The friction factor *m* between the billet and rolls was selected based on the correspondence of the sliding conditions and rolling time in the real process, and is equal to 0.95.

During RSR, the metal flow in the deformation zone occurs along a helicoidal trajectories of different lengths and it has a cyclic character. In the three-roll mill, the one cycle of deformation corresponds to the revolution of the billet by ⅓ of a turn or by 120°. In the deformation zone, the speeds of metal flow, their components, and the parameters of the helicoidal trajectories are changed [30].

After the calculation, the temperature field of the billet during RSR was analyzed, as well as the distribution of average normal stresses *σ_m_* and the distribution of effective (equivalent) strain *ε*_eff_ over the cross-section of the rod.

The average normal stress is defined as ⅓ of the sum of the stresses located on the main diagonal of the stress tensor (8):(8)$$\sigma_{m}=\frac{1}{3}\left(\sigma_{11}+\sigma_{22}+\sigma_{33}\right).$$

The complex characteristic of the strain rate is the intensity of the strain rates or the effective strain rate (9):(9)$$\dot{\bar{\varepsilon }}=\frac{\sqrt{2}}{3}\sqrt{{\left({\dot{\varepsilon }}_{1}-{\dot{\varepsilon }}_{2}\right)}^{2}+{\left({\dot{\varepsilon }}_{2}-{\dot{\varepsilon }}_{3}\right)}^{2}+{\left({\dot{\varepsilon }}_{3}-{\dot{\varepsilon }}_{1}\right)}^{2}}.$$

Based on the intensity of the strain rates, the effective strain can be calculated using Equation (10):(10)$$\varepsilon_{eff}=\int_{0}^{t}\dot{\bar{\varepsilon }}dt$$

## 3. Results and Discussion

### 3.1. FEM Simulation

A computer simulation of the RSR process will make it possible to analyze the thermal and deformation parameters and, as a result, determine their rational ranges for experimental rolling.

Based on the simulation results, the analysis of the temperature and stress–strain state of the rod in the process of RSR was carried out. Figure 4 shows the distribution of the temperature field of the rod in each pass. As can be seen, in the process of RSR, a complex formation of a temperature field in the volume of the rod occurs, which is associated with several factors. On the one hand, the billet is cooled in air and during contact with the rolls, which have a lower temperature. On the other hand, heat is released due to the work of deformation.

On the billet surface at the point of contact with the roll, a distinct area with a lower temperature (410–420 °C) can be seen. However, in the central part of the rod in the section of maximum reduction (section of the pinch of the rolls), there is an increase in temperature. Rolling at *T* = 450 °C with an average elongation ratio of 1.8 leads to heating of the central zone of the rod by an average of 20–30 °C. After leaving the deformation zone, the temperature of the rod decreases due to internal heat transfer and surface cooling.

Figure 5 shows the distribution of the effective strain in the cross-section of the rod after each pass. This parameter can be used to comprehensively characterize the effect of the deformation process on the material properties. According to previous studies [32,33], the RSR process achieves a high value of effective strain on the rod surface due to the severe shear deformations. In this case, a characteristic gradient distribution of strain across the radius of the rod can be seen. The maximum values are formed at the surface of the rod and uniformly decrease in the direction of the axial zone.

It should be noted that with an increase in the total elongation ratio, the effective strain increases not only near the surface, but also in the axial zone of the rod. Thus, the effective strain in the center after four passes is more than 10. The intense increase in strain may indicate a significant tendency of the alloy to develop deformations throughout the entire cross-section of the rod.

The value of the average stresses significantly affects the technological plasticity of the metals and alloys. The greater the negative average stresses, the greater the deformation without fracture is able to be achieved. Conversely, the fracture of the material is possible at high values of average stresses in combination with large effective strains.

Figure 6 shows the distribution of average stresses in the cross-section of the rod during the RSR process, which also reflects the gradient nature of the deformation during the RSR. In the axial zone of the rod, slight tensile stresses that have a positive value of 20–50 MPa are observed. Near the surface, at the area of contact with the work rolls, compressive stresses (from −150 to −200 MPa) are located.

In general, the simulation results show the possibility of obtaining rods from the ZCF alloy through the RSR method under the selected modes. The selected rolling temperature of 450 °C is optimal and ensures a uniform temperature of the rod after rolling in the range of 450–470 °C. Taking into account temperature heating during deformation, it is possible to deform the experimental alloy ZCF using the RSR method at heating temperatures from 420 °C with an elongation ratio up to 2.0 per pass. Given the fact that in the last passes the bar has a small diameter and a large length, its cooling rate will be much higher. In this regard, the speed parameters of the radial shear rolling must be corrected. The rotational speed of the rolls should be increased by 1.5–2 times relative to the first pass to compensate for heat losses.

### 3.2. Phase Composition Analysis

Before rolling, an analysis of the phase composition and structure of the initial ingot was carried out, which made it possible to assess the possibility of its deformation processing.

As can be seen from the liquidus surface of the Al–Zn–Mg–Ca–Fe system calculated at 8% Zn and 3.3% Mg (Figure 7a), the ZCF alloy falls into the hypereutectic region. Therefore, under equilibrium conditions, its crystallization should begin with the formation of the Al_3_Fe phase. However, the high cooling rate implemented in the EMC method leads to an expansion of the region of primary crystallization (Al) (dashed line in Figure 7a), thus the cast structure of the alloy is hypoeutectic (Figure 8a). In this case, the size of the dendritic cell (*d*) is ~7 µm, and Fe is bound into eutectic phase inclusions less than 3 µm in size (Figure 8b). According to the known relationship between the cooling rate (*V*) and the value of *d* [5], the estimated value of *V* (using the coefficients for the 7075 alloy [34]) is 600 K/s. In this case, needle-shaped particles characteristic of the Al_3_Fe phase [5] were absent. According to the morphological features, these inclusions may correspond to the Al_10_CaFe_2_ compound [16,18].

It is known that in alloys of the 7xxx series, during crystallization, zinc and magnesium are distributed between (Al) and the eutectic containing these elements [1]. According to the EMPA data, the concentrations of Zn and Mg (Al) are ~4.7 and ~1.9 wt.%, respectively, which are less than in the alloy (see in Table 1). The calculation of nonequilibrium crystallization according to the Sheil–Gulliver model shows the formation of the T (Al_2_Mg_3_Zn_3_) phase enriched in Zn and Mg at 480 °C (Figure 7b).

Annealing at 450 °C for 3 h leads to complete dissolution of Mg in (Al), while the concentration of Zn in (Al) is slightly lower than in the alloy (see Table 1). The latter is due to the previously revealed fact [19,20] that Zn also tends to dissolve in the (Al, Zn)_4_Ca phase. Nevertheless, during annealing, the nonequilibrium T-phase completely dissolves in the aluminum matrix. This is consistent with the isothermal section of the Al–Zn–Mg–Fe–Ni system calculated at 0.8% Ca, 1.1% Fe, and 450 °C, as the point corresponding to the composition of the experimental alloy falls into the boundary between two-phase fields with and without the T-phase. However, considering the lower concentration of the Zn in (Al), in the actual case the composition point of the alloy should move (marked by the arrow in diagram, Figure 7c) to the T-phase free field (TTAL database does not contain proper data on the solubility of Zn in the (Al, Zn)_4_Ca phase, so the obtained calculated data can only be used for the assessments). At the same time, the morphology of eutectic inclusions of the Fe-containing phase changes insignificantly (Figure 8c,d). As this phase region exists in a wide temperature range, namely between the temperatures of solvus (450 °C) and equilibrium solidus (531 °C), heat treatment was also carried out in a two-stage mode: 450 °C, 3 h + 500 °C, 3 h.

As can be seen from Figure 8e,f, this mode led to the spheroidization of Fe-containing particles. According to previous works on alloys of the Al–Zn–Mg–Ca–Fe system [19,20], it is precisely this structure that makes it possible to provide a high deformation plasticity. However, based on the fact that additional deformation heating of the billet occurs during the RSR process (which can lead to partial melting), heating at 450 °C was chosen. This temperature corresponds to the rolling temperature (selected from the simulation results); thus, homogenization was combined with heating before RSR.

### 3.3. Experimental RSR

The RSR of the rod according to the selected mode (see Table 2) showed that the deformation plasticity of the experimental alloy containing more than 1% Fe was sufficient to obtain a rod with a total elongation ratio of 10.5 (logarithmic strain 2.35). There were no visible defects on the rod, both outside (see Figure 2b) and inside. In the process of deformation, a structure characteristic of composites was formed. Globular particles of the Al_10_CaFe_2_ phase were uniformly distributed in the aluminum matrix (Figure 9). The presence of a fibrous structure was detected only in the center of the rod in the longitudinal direction (Figure 9a), which is typical for RSR [25,26,28].

To estimate the degree of hardening in different zones of the rod processed by RSR, the effect of aging time at 150 °C on the hardness (after quenching from 450 °C) was studied. It was found that after 3 h of exposure, the hardness in the center of the rod was approximately 20 HV lower than in the remaining zones (Figure 10). However, after 6 h, the hardness values almost levelled out (195–200 HV).

An analysis of the peak aged structure performed using TEM revealed the precipitate structure with a very high number density (Figure 11a) and with an average size of individual precipitates of about 5 nm in a diameter (Figure 11b). The experimental diffraction patterns are good evidence that the observed precipitates belong to the *ƞ*(MgZn_2_)-phase (Figure 11c), which agrees well with known data [6,7,8,35,36]. The detected precipitation structure explained the observed high peak aged hardness of the billet well.

## 4. Conclusions

(1)Based on computational (Thermo-Calc) and experimental (SEM, EMPA, TEM, and DSC) methods, the effect of radial shear rolling (RSR) on the structure and hardening of the Al–8%Zn–3.3%Mg–0.8%-Ca–1.1%Fe alloy obtained by casting into an electromagnetic mold (EMC) in the form of an ingot with a diameter of 33 mm was studied.(2)It was shown that the EMC method makes it possible to obtain a highly dispersed structure in the cast state due to the high solidification rate (600 K/s); the size of the dendritic cell was ~7 μm while the entire amount of Fe was bound into eutectic Fe-containing inclusions with an average size of less than 3 µm.(3)An analysis of the RSR parameters using the FEM simulation showed that the selected modes made it possible to obtain rods with a total elongation ratio more than 10. The selected rolling temperature of 450 °C was optimal and provided a uniform temperature of the rod after RSR in the range of 450–470 °C. The investigated alloy had a significant tendency to develop effective strains over the entire cross-section of the rod.(4)Heating according to the mode of 450 °C for 3 h led to complete dissolution of the nonequilibrium brittle Zn and Mg-containing phases in the aluminum solid solution. This structure provides a high deformation ductility at RSR and makes it possible to obtain rods without defects and with a diameter of 10 mm.(5)In the process of deformation, a structure is formed that is characteristic of composites; globular particles of the Ca and Fe-containing phases are uniformly distributed in the aluminum matrix. The flow features during RSR and the developed shear strains near the surface of the rod provide refining and a decrease in the size of the initial particles of the cast origin phase.(6)The effect of aging temperature (after quenching) on the hardness of the experimental alloy was studied. It was shown that the maximum hardness (195–200 HV) in all of the zones was observed after 6 h of aging at 150 °C due to the formation of secondary precipitates with an average characteristic size of ~5 nm and a high distribution density.

## Figures and Tables

**Figure 1 materials-16-00677-f001:**
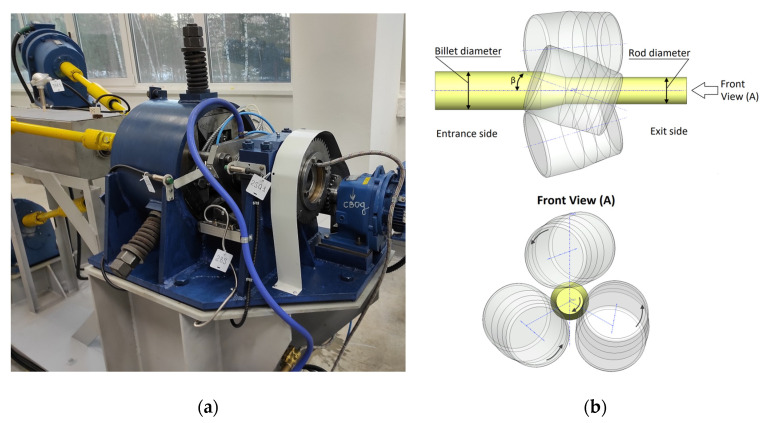
RSR Minimill “MISIS 14–40” (**a**) and schematic diagram of RSR (**b**).

**Figure 2 materials-16-00677-f002:**
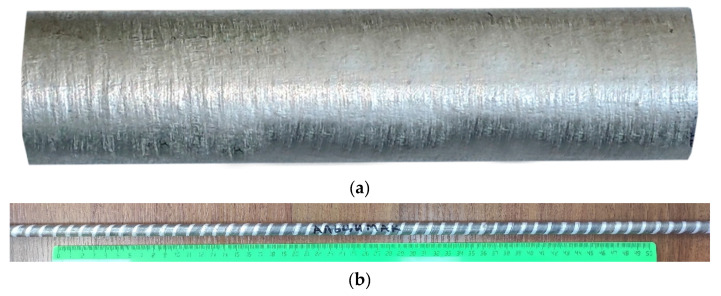
Ingot of the experimental alloy manufactured by EMC (**a**) and the rod processed by RSR from the EMC ingot (**b**).

**Figure 3 materials-16-00677-f003:**
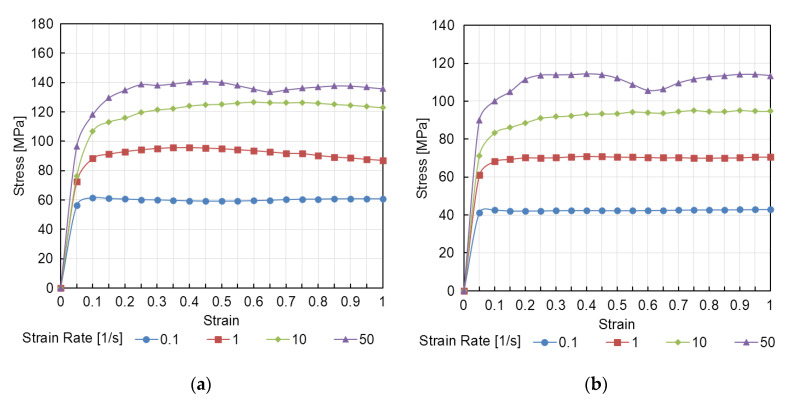
Plastic flow curves for the ZCF aluminum alloy for temperatures of (**a**) 400 °C, (**b**) 450 °C, and (**c**) 500 °C.

**Figure 4 materials-16-00677-f004:**
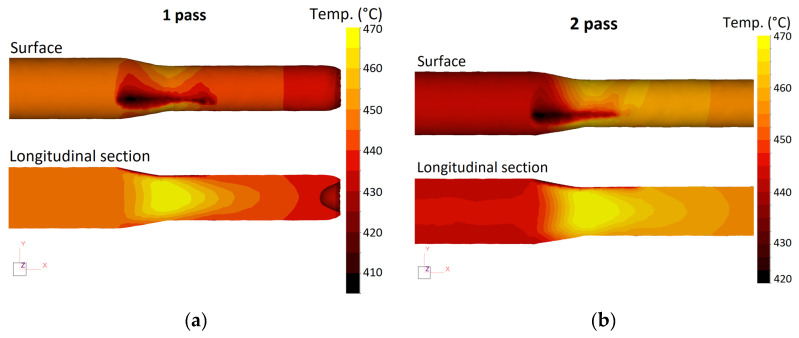
Distribution of the temperature field of the billet in the deformation zone in each pass: (**a**) first pass; (**b**) second pass; (**c**) third pass; (**d**) fourth pass.

**Figure 5 materials-16-00677-f005:**
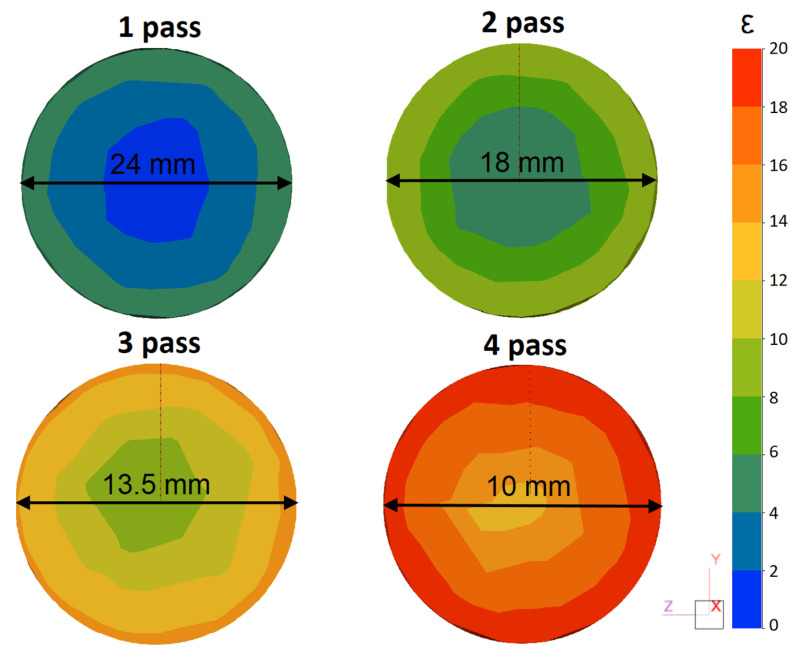
Distribution of effective strain over the cross-section of the rod after each pass.

**Figure 6 materials-16-00677-f006:**
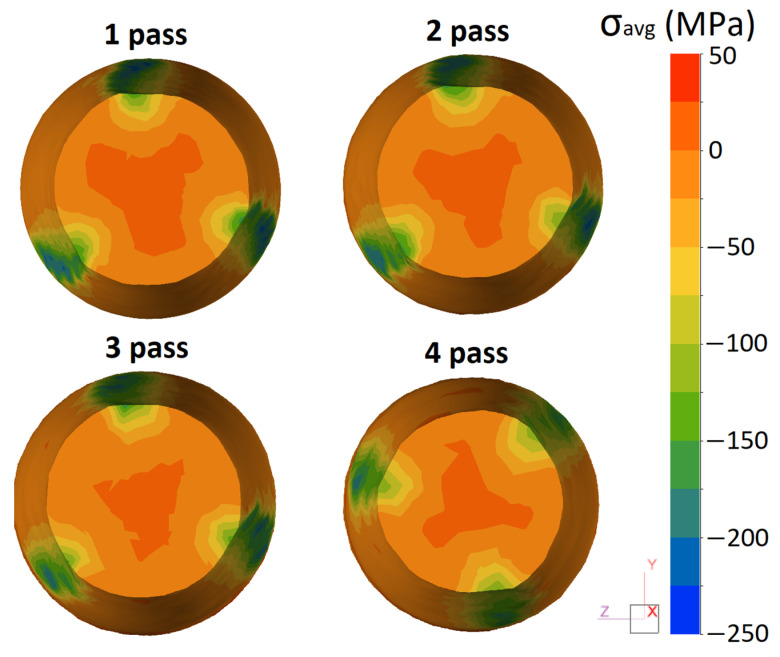
Distribution of average stresses over the cross-section of the rod in each pass (cross-section of roll pinch).

**Figure 7 materials-16-00677-f007:**
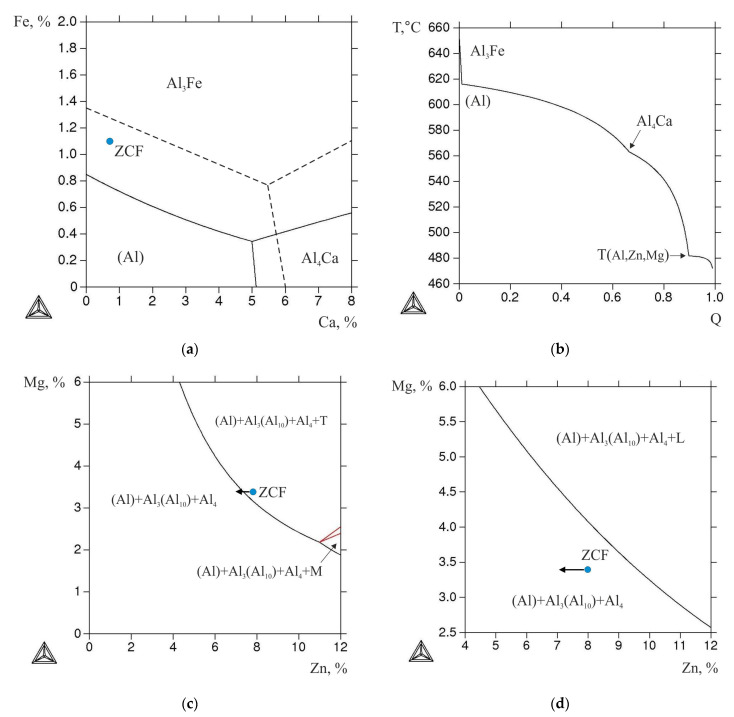
Calculated fragments of Al–Zn–Mg–Ca–Fe phase diagram (position of experimental alloy CZF is shown): (**a**) liquidus projection at 8% Zn and 3.3% Mg; (**b**) dependence of mass fraction of solid phases (Q) versus temperature (Sheil–Gulliver simulation) for alloy ZCF; (**c**) isothermal section at 0.8% Ca, 1.1% Fe, and 450 °C; (**d**) isothermal section at 0.8% Ca, 1.1% Fe, and 500 °C.

**Figure 8 materials-16-00677-f008:**
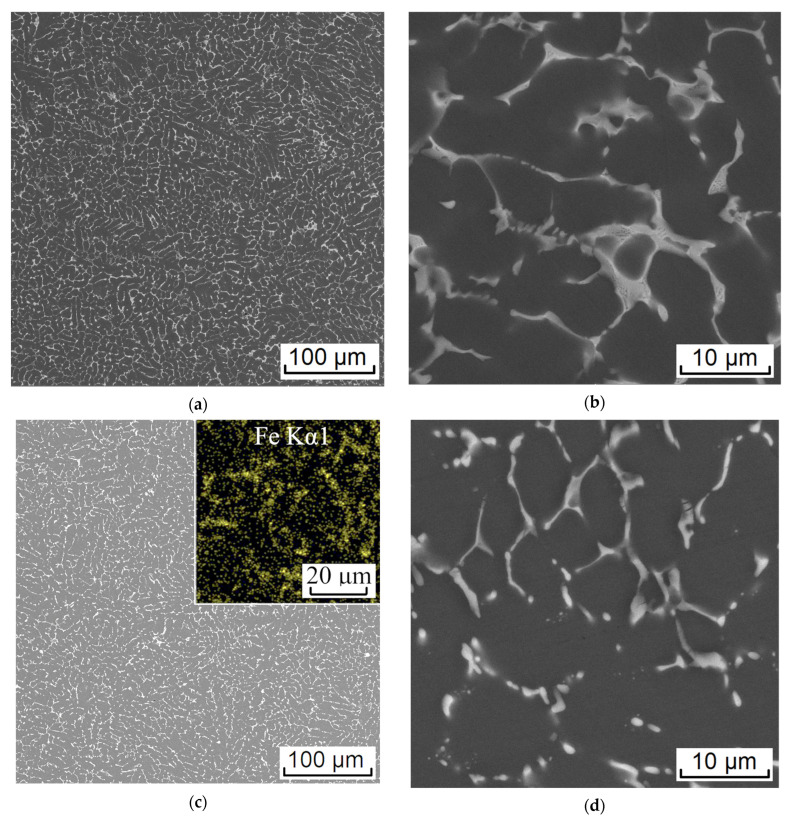
Microstructure (SEM) of the experimental alloy ZCF in the as-cast state (**a**,**b**) and after annealing at 450 °C (**c**,**d**) and 500 °C (**e**,**f**).

**Figure 9 materials-16-00677-f009:**
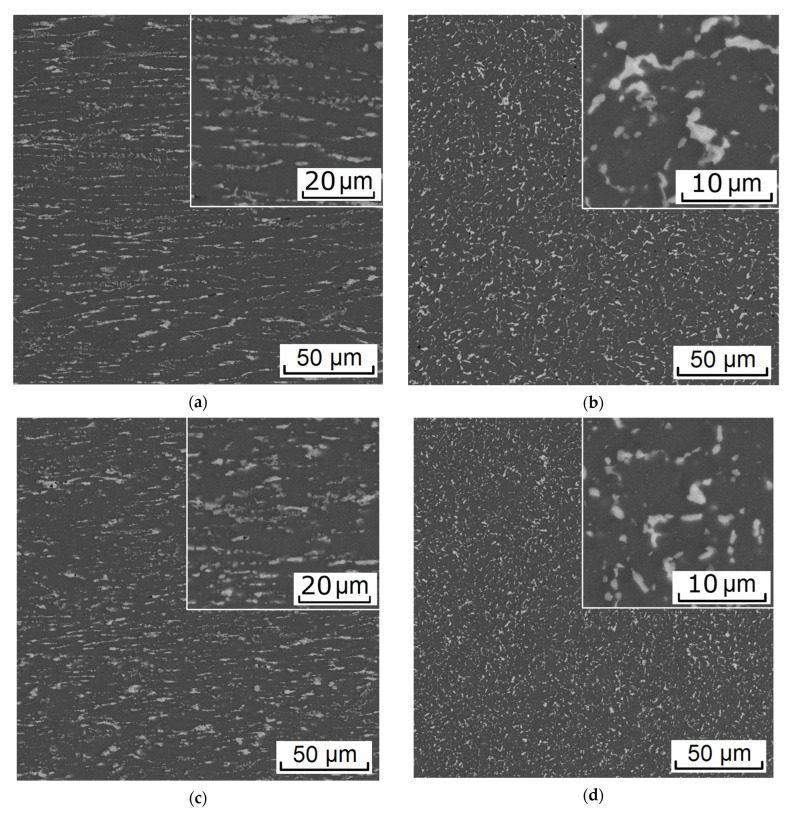
Microstructure (SEM) of the experimental alloy processed RSR: (**a**,**b**) center, (**c**,**d**) ½ radius, and (**e**,**f**) edge. (**a**,**c**,**e**) Longitudinal section and (**b**,**d**,**f**) transverse section.

**Figure 10 materials-16-00677-f010:**
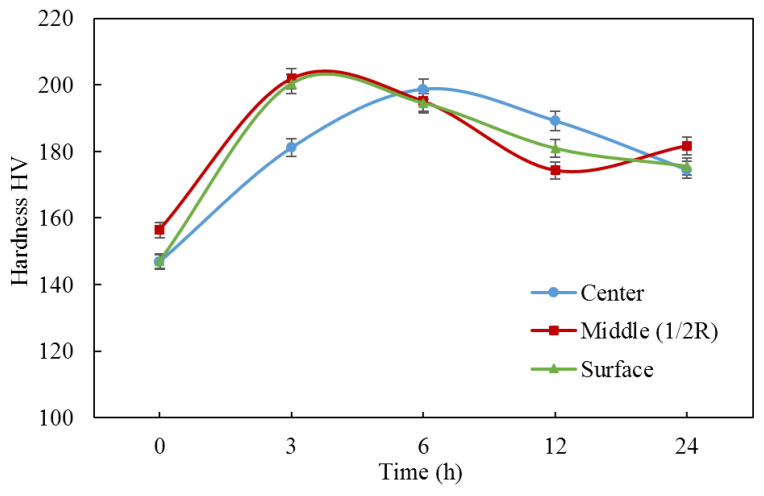
Hardness of experimental alloy (RSR) vs. aging time at 150 °C (solution treatment at 450 °C, 1 h + water cooling).

**Figure 11 materials-16-00677-f011:**
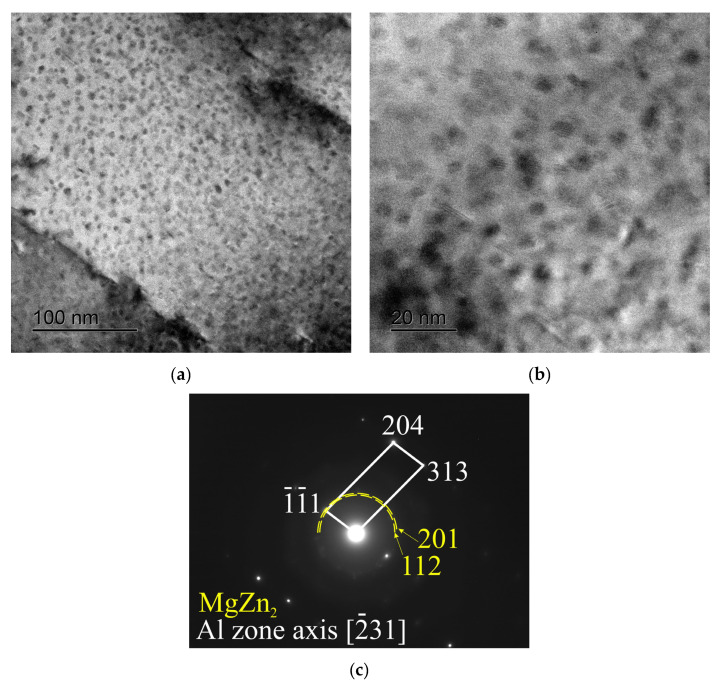
TEM structure of the experimental alloy after solution treatment and aging (150 °C, 6 h): (**a**) dark field, (**b**) high resolution image, and (**c**) selected area diffraction patterns (SADP) taken along the [−231] Al zone axis.

**Table 1 materials-16-00677-t001:** Chemical composition of the experimental aluminum alloy and aluminum solid solution (Al).

Alloy		Concentration, wt.%				
		Zn	Mg	Ca	Fe	Si
ZCF	Nominal	8.0	3.3	0.8	1.1	<0.1
	Actual	7.84 ± 0.12	3.23 ± 0.10	0.79 ± 0.02	1.12 ± 0.09	<0.01
(Al)	As-cast	4.72 ± 0.16	1.89 ± 0.07	0.02 ± 0.02	0.08 ± 0.05	<0.01
	quenching from 450 °C	6.95 ± 0.11	3.73 ± 0.03	0.04 ± 0.03	0.15 ± 0.08	<0.01

**Table 2 materials-16-00677-t002:** Modes of radial shear rolling.

Pass No.	Billet Diameter (mm)	Rod Diameter (mm)	Elongation Ratio	Total Elongation Ratio	Temperature (°C)	Roll Rotary Velocity (rpm)
1	33	24	1.89	1.89	450	50
2	24	18	1.78	3.36	450	50
3	18	13.5	1.78	5.98	450	50
4	13.5	10	1.75	10.47	450	90

## Data Availability

The data presented in this study are available on request from the corresponding author.

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
