# Peer review of "Effect of Radial-Shear Rolling on the Structure and Hardening of an Al–8%Zn–3.3%Mg–0.8%Ca–1.1%Fe Alloy Manufactured by Electromagnetic Casting"

_materials, 2023, doi:10.3390/ma16020677_

Round 1
Reviewer 1 Report
The subject of research presented in the paper is an effect of radial-shear rolling on structure and hardening of Al–8%Zn–3.3%Mg–0.8%Ca–1.1%Fe alloy manufactured by electromagnetic casting. In this study, the effect of radial shear rolling on the formation of the structure and hardening of the alloy obtained by the electromagnetic casting method were studied.
The authors concentrated on the computer simulation of the radial shear rolling process made it possible to analyse the thermal and deformation parameters, and an analysis of the phase composition and structure of the initial ingot was carried out and analysis of the material after rolling. The article submitted for review has a high scientific and practical value.
The work presented for review is very interesting, deals with an interesting topic and fits well with the scope of the journal. The research methodology presented is good. However, the desirability of using TEM in these studies should be discussed more and what phases are visible on the microstructure should be indicated. I also miss EDX analysis of phases and spectral peaks.
I'm think the paper contains limited discussions, especially since there is a lack of comparison to the results of other authors, therefore a significant discussion regarding the underlying mechanisms controlling the observed findings should be realized.
I'm found an interesting publication regarding the discussed research methodology https://www.mdpi.com/1996-1944/15/3/954
Author Response
Dear Reviewer,
We appreciate your detailed comments. They raised new topics that enriched the content of the paper. Our responses are marked in Italics below and the changes in the manuscript are marked using yellow (regarding technical correction) and blue (regarding writing and grammar corrections) highlighting.
Response: Thanks for the comments.
- The results of EDX analysis are added in Table 1 (for aluminum solid solution) and in Fig.8c (for Fe mapping).
- To identify the observed precipitates, the selected area diffraction patterns taken along [-231]Al zone axis were also obtained and analyzed. According to the data obtained, the precipitates belong to the ƞ(MgZn2)-phase, which agrees well with known data. Corresponding explanations are added in the article.
- The discussion and analysis presented in almost all parts of the article have been substantially expanded.

Reviewer 2 Report
I can recommend the publication of this manuscript after a minor revision.
1. Write keywords in alphabetical order.
2. Insert references for all mathematical formulas.
3. The writing must be improved, it is still very poor with numerous wrong terms, typos, or grammar mistakes.
4. Page 2. ........[5-10], ....[23-28].... You will likely need to re-write your citation sentences, rather than simply replacing the numbers with Authors’ names. This is due to the fact that in order to give readers the maximum appreciation of how your work builds on previous results, each one of the cited sources should be discussed individually and explicitly to demonstrate their significance to your study. We ask that you use the authors' surnames as the subject of a verb, and then state in one or two sentences what they claim, what evidence they provide to support their claim, and how you evaluate their work. We also, therefore, ask that you avoid citing more than one reference in one sentence. This will give you a chance to discuss each reference separately.
What we are asking for is something like this: “Smith (2011) describes the development of a finite element model of hot forging and claims excellent agreement between the model and experiments. A much more detailed comparison would be required to evaluate the precise conditions under which finite element modeling is truly accurate."
5. How many samples were used in the experimental method?
6. The parameters of the model for the simulation of the RSR process were chosen by an optimal procedure, or for other reasons? Explain and give more details about this procedure.
7. If possible, can you insert a mathematical model (some mathematical relations with the process parameters) of this kinematic process?
8. Give more details about the statistical analysis applied in this manuscript.
9. Re-write the corresponding theory associated with equations 1 to 3, and give the corresponding theoretical details. Extend this section with particular mathematical formulas used in this study.
10. Specify the errors that are included in these numerical simulations for all parameters.
11. Specify the limits of this study.
12. Explain with more details sentences from pages:
a) 9 “.... In the last passes, the rotational speed of the rolls should be increased by 1.5-2 times relative to the first pass to compensate for heat losses.”
b) 15. Section Conclusion: (3) and (5).
13. References are not written according to the Guide of authors (i.e. sometimes are used: https://doi.org/..., doi:... and so on)
14. If possible, authors may consider citing the following references:
[1] Langdon, T.G. The characteristics of grain refinement in materials processed by severe plastic deformation. Rev. Adv. Mater. Sci. 2006, 13, 6–14
[2] Ş. Ţălu, Micro and nanoscale characterization of three-dimensional surfaces. Basics and applications. Napoca Star Publishing House, Cluj-Napoca, Romania, 2015.
This paper can be published after the mentioned revisions.
Author Response
Dear Reviewer,
We appreciate your detailed comments. They raised new topics that enriched the content of the paper. Our responses are marked in Italics below, and the changes in the manuscript are marked using yellow (regarding technical correction) and blue (regarding writing and grammar corrections) highlighting.
- Write keywords in alphabetical order.
Response: Thank you for the comment. We have corrected the order of keywords according to your recommendations.
- Insert references for all mathematical formulas.
Response: Thank you for your comment. We have added the references for all formulas.
- The writing must be improved, it is still very poor with numerous wrong terms, typos, or grammar mistakes.
Response: Thank you for the comment. We have checked the text again and corrected typos and grammar errors. All the main terms that we used to describe the results are generally accepted in the technical literature on the theory of metal forming and materials science.
- Page 2. ........[5-10], ....[23-28].... You will likely need to re-write your citation sentences, rather than simply replacing the numbers with Authors’ names. This is due to the fact that in order to give readers the maximum appreciation of how your work builds on previous results, each one of the cited sources should be discussed individually and explicitly to demonstrate their significance to your study. We ask that you use the authors' surnames as the subject of a verb, and then state in one or two sentences what they claim, what evidence they provide to support their claim, and how you evaluate their work. We also, therefore, ask that you avoid citing more than one reference in one sentence. This will give you a chance to discuss each reference separately.
What we are asking for is something like this: “Smith (2011) describes the development of a finite element model of hot forging and claims excellent agreement between the model and experiments. A much more detailed comparison would be required to evaluate the precise conditions under which finite element modeling is truly accurate."
Response: We re-write all the sentences with citations.
- How many samples were used in the experimental method?
Response: The 3 samples cut from RSR rod were used.
- The parameters of the model for the simulation of the RSR process were chosen by an optimal procedure, or for other reasons? Explain and give more details about this procedure.
Response: The parameters of model are corresponded with the conditions of universal method of RSR for deformation of solid workpieces from different metals and alloys [see 24, 25 references in the article for more details]. The main initial parameters of the model are chosen based on their correspondence to the experimental parameters of real RSR process.
- If possible, can you insert a mathematical model (some mathematical relations with the process parameters) of this kinematic process?
Response: We have added additional mathematical relations with the process parameters (regarding elongation ratio and temperature calculations). The kinematic model of the RSR process was considered in detail in a previous article by the authors Galkin, Gamin et al. (reference 30 in the article), so we added a brief description and provided a citation to this work.
- Give more details about the statistical analysis applied in this manuscript.
Response: The Vickers hardness was measured at least five times at each point. The standard deviation doesn’t exceed 3 HV and marked on the Fig.10. The standard deviations for chemical composition are given in Table 1
- Re-write the corresponding theory associated with equations 1 to 3, and give the corresponding theoretical details. Extend this section with particular mathematical formulas used in this study.
Response: Thank you for your comment. We have added additional mathematical relations with the process parameters (regarding elongation ratio and temperature calculations).
- Specify the errors that are included in these numerical simulations for all parameters.
Response: As practice shows, the relative error of equations (1-6) does not exceed 10 %. The relative error of FEM simulation results depends on many factors such as the elements size of finite element mesh, its method of fragmentation, the assumption of distribution uniformity of initial temperature, etc. As the earlier comparative analysis of the model and the real experiment shows, this error does not exceed 7-10 %.
- Specify the limits of this study.
Response: In this study, the effect of RSR on Al–Zn–Mg–Ca–Fe alloy manufactured by electromagnetic casting was carried out. The aim of this work was to study the main stress-strain parameters of the RSR process on the structure evolution for the experimental alloy of Al–Zn–Mg–Ca–Fe system.
- Explain with more details sentences from pages:
- a) 9 “.... In the last passes, the rotational speed of the rolls should be increased by 1.5-2 times relative to the first pass to compensate for heat losses.”
- b) 15. Section Conclusion: (3) and (5).
Response: We have expanded the description and made the conclusions more specific.
- References are not written according to the Guide of Authors (i.e. sometimes are used: https://doi.org/..., doi:... and so on)
Response: Thanks for the comment. We have corrected the formatting of references in accordance with the Guide of Authors.
- If possible, authors may consider citing the following references:
[1] Langdon, T.G. The characteristics of grain refinement in materials processed by severe plastic deformation. Rev. Adv. Mater. Sci. 2006, 13, 6–14
[2] Ş. Ţălu, Micro and nanoscale characterization of three-dimensional surfaces. Basics and applications. Napoca Star Publishing House, Cluj-Napoca, Romania, 2015.
Response: Thanks for the recommendation. We have added the first reference that you recommended.

Reviewer 3 Report
In this article, experiments and simulations were adopted to research the influence of RSR on the formation of the structure and hardening of the Al-Zn-Mg-Ca-Fe alloy. It is an interesting field, but the following issues should be noted:
1. The authors must double-check the article's writing.
(a) For example: In the Introduction, “Aluminum alloys are one of the most common structural materials. This is due to both its….”.
“This is” refers to alloys. Please make sure the correct singular-plural pair.
(b) In Fig. 6, there is an additional (b). “(d) isothermal section at 0.8%Ca, 1.1%Fe and 500 0С (b)”
2. The workpiece is ZCF alloy. Therefore, the authors should supplement the basic mechanical properties parameters of ZCF alloy in FEM simulation. Meanwhile, the authors also need to provide the constitutive equation parameters of ZCF alloy.
3. In Fig. 8, the authors researched the microstructure of processed alloys. If possible, the authors should provide partially enlarged drawings to make them clear for authors (likes Figs. 7b, d, f).
4. The authors need to adjust the sequence of conclusions properly.
For example, the conclusion involving FEM simulation should be No.2, which is consistent with the above content.
Author Response
Dear Reviewer,
We appreciate your detailed comments. They raised new topics that enriched the content of the paper. Our responses are marked in Italics below, and the changes in the manuscript are marked using yellow (regarding technical correction) and blue (regarding writing and grammar corrections) highlighting.
- The authors must double-check the article's writing.
(a) For example: In the Introduction, “Aluminum alloys are one of the most common structural materials. This is due to both its….”.
“This is” refers to alloys. Please make sure the correct singular-plural pair.
(b) In Fig. 6, there is an additional (b). “(d) isothermal section at 0.8%Ca, 1.1%Fe and 500 0С (b)”
Response: Thank you for the comment. We have corrected the mistakes.
- The workpiece is ZCF alloy. Therefore, the authors should supplement the basic mechanical properties parameters of ZCF alloy in FEM simulation. Meanwhile, the authors also need to provide the constitutive equation parameters of ZCF alloy.
Response: The rheological properties of ZCF alloy, which were used in FEM simulation, have been added to the article (Fig. 3 in the article).
- In Fig. 8, the authors researched the microstructure of processed alloys. If possible, the authors should provide partially enlarged drawings to make them clear for authors (likes Figs. 7b, d, f).
Response: We have added the enlarged microstructures on Fig. 9.
- The authors need to adjust the sequence of conclusions properly.
For example, the conclusion involving FEM simulation should be No.2, which is consistent with the above content.
Response: Thank you for the comment. We have corrected the structure of the conclusion section. In particular, we swapped conclusions (3) and (4), which, in our opinion, better reflects the structure of the results obtained.

Round 2
Reviewer 1 Report
The authors took into account the reviewer's suggestions